# OpenReview forum: "The Encoding-Behavior Dissociation: How Distributed Safety Representations Yield Single-Direction Vulnerabilities in Vision-Language Model"
_TMLR — Withdrawn by Authors_

### Review · Reviewer_jAQd · 2026-05-26

**Summary Of Contributions:**

* Identifies the “Encoding-Behavior Dissociation” phenomenon, revealing a mismatch between high-dimensional safety encoding and one-dimensional behavioral refusal in VLMs.
* Demonstrates that VLM safety encoding is highly robust, distributed across over 50 PCA components, all token positions, and all attention heads.
* Causally verifies that VLM refusal behavior relies on a vulnerable bottleneck governed by a single threshold along the dominant DIM direction.
* Proposes a novel geometry-aware adversarial PGD attack that targets this geometric bottleneck to shift internal representations into the benign domain.
*Uncovers how visual context dynamically modulates safety, where safety-relevant images enhance text separation while neutral images suppress it.

**Audience:**

Yes

**Audience Explanation:**

Addresses a timely topic of interest to the TMLR audience, particularly those working on multimodal security, AI alignment, representation engineering, and mechanistic interpretability.

**Claims And Evidence:**

Yes

**Claims Explanation:**

* Evaluates various open-source VLMs (e.g., PaliGemma 2, Qwen2-VL, LLaVA-v1.6, Phi-3.5-Vision) across scales from 2B to 7B parameters to prove a consistent vulnerability geometry.
* Establishes a causal link to the refusal circuit through bidirectional activation steering experiments, moving beyond mere correlation by demonstrating both refusal induction and suppression via sign inversion.
* Validates the theoretical hypotheses through multi-faceted analyses, including AUROC scaling over PCA components, token-position/attention-head attributions, and gradient-based token attribution diffusion.
* Conducts cross-validation using an independent LLM judge (Gemini 3.1 Flash) to complement the keyword classifier, achieving a Cohen's kappa of 0.784 to ensure high reliability.

**Requested Changes:**

* The high-dimensional encoding analysis is primarily conducted on pretrained PaliGemma 2, whereas the 1D behavioral readout and attacks are evaluated on separate, instruction-tuned models like Qwen, LLaVA, and Phi. This disconnect breaks the causal chain needed to prove the dissociation within a single, consistent architecture.
* The assessment relies heavily on a keyword-based classifier, where the mere absence of refusal keywords does not guarantee actionable compliance. This is a critical distinction since steering can degrade generation quality and shorten responses, meaning "refusal bypass" may be conflated with output corruption.
* The observed high PCA dimensionality might capture dataset artifacts, text-image congruence, or stylistic disparities rather than pure safety representations. The separation between datasets (e.g., SPA-VL vs. VQAv2) could simply reflect a domain classifier, requiring stronger matched controls to definitively isolate the safety subspace.

---

### Review · Reviewer_CETA · 2026-05-28

**Summary Of Contributions:**

The authors set out to explore how safety is implemented in VLMs compared to LLMs.

**Additional Comments:**

real concern about the reliability of the article, in particular, the references.

**Audience:**

Yes

**Audience Explanation:**

Yes, this is a very important topic regarding society and the impact of AI deployment

**Claims And Evidence:**

Yes

**Claims Explanation:**

yes, just a note about dissociation. The "dissociation" may be an artifact of the evaluation design
The paper compares encoding robustness (measured by iterative DIM ablation on PaliGemma 2) with behavioral fragility (measured by PGD attacks on instruction-tuned models). These are fundamentally different interventions on fundamentally different models. Iterative ablation modifies the internal representation; PGD attacks optimize input pixels. Demonstrating that one intervention fails and another succeeds on different models does not constitute evidence of a dissociation — it is a comparison of apples and oranges. A rigorous test would require showing that iterative ablation of the safety subspace fails to suppress refusal in instruction-tuned VLMs, while PGD attacks succeed, on the same model. Appendix C.1 partially addresses this for Qwen2-VL-2B-Instruct, but the results there are actually more nuanced than acknowledged: ablating 1 direction reduces refusal from 66% to 24%, which is quite substantial and partially contradicts the "robust encoding" narrative.

**Requested Changes:**

- the abbreviation PGD is not defined
- there might be issues with references: I failed to find "Shang Gou et al. Cross-modal refusal mitigation in vision-language models. 2025.", the first name for "Alexander Korznikov et al. Rogue scalpel: Random activation perturbations erode LLM safety. arXiv
preprint, 2025." is wrong, and the reference is incomplete, the title of "Jiwoo Lee et al. CAST: Conditional activation steering for selective refusal. arXiv preprint, 2025." is false, and the first name is false too, citation "Yifan Li et al. Images are Achilles’ heel of alignment: Exploiting visual vulnerabilities for jailbreaking multimodal large language models. arXiv preprint, 2024." is incomplete, reference "George Pantazopoulos et al. Learning to forget: Multimodal models are more jailbreak-susceptible than their backbones. arXiv preprint, 2024." is false, the first name is inaccurate, and the actual title is "Learning To See But Forgetting To Follow: Visual Instruction Tuning Makes LLMs More Prone To Jailbreak Attacks", could not find "Jialiang Tu et al. More than one direction: Disentangling multiple refusal directions in LLMs. arXiv preprint, 2026.", also did not find "Wei Xiong et al. Steering vectors unintentionally erode safety in LLMs. arXiv preprint, 2026.", also not "Zhiyuan Ying et al. BAP: Better adversarial perturbation via joint image-text optimization. arXiv preprint arXiv:2406.04031, 2024.", mixed with "Ying, Zonghao, Aishan Liu, Tianyuan Zhang, Zhengmin Yu, Siyuan Liang, Xianglong Liu, and Dacheng Tao. "Jailbreak vision language models via bi-modal adversarial prompt." IEEE Transactions on Information Forensics and Security (2025)." (different first name of the first author, different title, but right arxiv number)
- the paper is very difficult and technical to read, a more gentle introduction, and additional figures to explain the different attacks, and modes "controlled vision" and "neutral images". For instance, the figures proposed in https://arxiv.org/html/2605.00583v1 are very illustrative. Additionnally, I might be missing something, but I don't understand whether neutral images are adversarial or just very safe images. The sentence "Neutral images: Pairs using VQAv2 everyday images, testing whether non-safety visual content modulates the text safety signal." is misleading on that, as the meaning of "non-safety visual content" is not clear. Please rephrase to make it clear that the images are benign
- all the models have a small number of parameters, how about the other works in the field? how could it impact the results? it should be discussed in the limitations paragraph if it is not possible to test with bigger models
- Several cross-references in the paper appear as "??"
- figure 9 : no tokens in bold or bold-italic
- The paper does not sufficiently distinguish its theoretical contribution from prior work. Also there lacks some insights about why the behavior of VLMs is different from LLMs, besides having a more complex encoding.
- The encoding geometry analysis is conducted exclusively on PaliGemma 2 (pretrained, 0% refusal), while the behavioral and attack experiments are conducted on different models (Qwen2-VL, LLaVA, Phi).

So overall, the paper needs a lot of work to rewrite it in a way that makes the contributions easier to understand, correction of the numerous formatting errors, that really alter the confidence one can have in the article.

---

### Review · Reviewer_BFEj · 2026-06-15

**Summary Of Contributions:**

**Summary.** The paper studies the geometry of safety refusal in vision-language models. It argues that, unlike text-only LLMs, safety information in VLMs is encoded in a distributed, high-dimensional manner across many directions, token positions, and attention heads. Despite this, refusal behavior in instruction-tuned VLMs can be controlled through a dominant Difference-in-Means direction. Building on this, the paper develops geometry-aware adversarial image attacks that achieve high refusal-bypass rates across several VLM architectures. The authors summarize these findings as an “encoding–behavior dissociation”: safety representations appear robust at the encoding level but remain fragile at the behavioral level.

**Strengths.**
The paper has a clear central story—VLM safety representations can be distributed at the encoding level, while refusal behavior can still depend on a low-dimensional steering direction—that I found interesting and well supported by experiments.

**Weaknesses.**
- There is no mitigation discussion. Since the paper identifies a safety vulnerability, I would expect a discussion of possible defenses or design implications.
- The main results (Section 4) could be organized more clearly (see requested changes below).

**Additional Comments:**

Some questions:
- [10] Could you clarify the relationship between the PCA dimensionality results in Figure 5 and the iterative DIM ablation results in Figure 6? The text-only PG-2 condition appears highly concentrated in Figure 5 but robust to direction ablation in Figure 6.
- [11] In Table 8, is the reported 30.8% bypass for Phi-3.5-VI under $\varepsilon=16/255$ accurate? In general, why are the bypass rates not monotonic in the perturbation budget?
- [12] The text says 4-bit quantized PGD achieves 50% bypass, but Table 8 reports roughly 95–97%.

**Audience:**

Yes

**Audience Explanation:**

Yes. The paper studies whether refusal behavior is mechanistically fragile despite distributed safety encoding. I expect this to be of interest to researchers working on multimodal safety, mechanistic interpretability, and alignment.

**Broader Impact Concerns:**

The paper identifies a safety vulnerability in instruction-tuned VLMs and demonstrates successful adversarial attacks. However, it does not discuss the defensive implications of these findings. Given the nature of the results, a mitigation discussion should be added before publication. (The current Broader Impact section contains an unresolved reference to such a discussion, which I was unable to find elsewhere in the paper.)

**Claims And Evidence:**

Yes

**Claims Explanation:**

The paper makes three main claims, which I find well supported:

- VLM safety information is more distributed at the encoding level than in text-only LLMs. This is supported by the PCA analysis in Fig 5 and Table 3, and by the iterative ablation results (Table 4, Fig 6).
- Instruction-tuned VLM refusal behavior can still be controlled through a dominant DIM-like direction. This is supported by the activation-steering results (Tables 6—7, Fig 7).
- The behavioral bottleneck can be exploited by geometry-aware adversarial image attacks. This is supported by the PGD bypass results in Table 8 and Fig 8, the scaling comparison in Tables 9–10, and the cross-model transfer results in Table 12.

**Requested Changes:**

- [1, required] Add a mitigation discussion. Since the paper identifies a safety vulnerability, it should discuss plausible mitigation directions, even if it does not implement a full defense.
- [2, required] Report uncertainty for attack results (mean p.m. std over multiple seeds), see e.g. Table 8.

Exposition/clarity:
- [3] Readers who are less familiar with the safety and mechanistic interpretability literature may benefit from a slightly more detailed methodology section. In particular, the definition of DIM (3.1) and activation steering (3.4) could be expanded. Also, the perturbation budgets should be defined in the "Per-image adversarial PGD" paragraph in 3.4.
- [4] The paper would be easier to follow if the findings were grouped into subsections around the main argument—for example: encoding geometry, behavioral control, attacks, and then generalization and limitations. Right now, the results section reads as a sequence of experiments rather than an argument leading to the paper’s main claim.
- [5] I'd move token attribution to appendix or label it as supporting analysis. It is useful but less central than PCA, ablation, steering, and PGD.
- [6] What's the point of response length in Table 6, Fig 7? If just a generation-collapse diagnostic, I would just briefly mention in text, as it dilutes the primary behavior metric (refusal rate).

Some typos:
- [7] DIM acronym in page 2 is defined on page 3. PGD is not defined.
- [8] There's a repetition in the activation-steering section (duplicated signed-$\alpha$ explanation).
- [9] Fix broken reference on page 8.

---

### Note · Authors · 2026-07-13

**Comment:**

Dear AE and Reviewers,

We thank you for your time in processing our submission and providing constructive feedback on the paper.

Due to computational constraints, we cannot proceed to complete the experiments required for an appropriate revision, and we therefore regret that we must withdraw our submission.

We sincerely thank you for your time and effort.

Best regards, \
The Authors

**Withdrawal Confirmation:**

I have read and agree with the venue's withdrawal policy on behalf of myself and my co-authors.